# Peptide AEDL Activates Metabolism and Autophagy in Root Cells of *Nicotiana tabacum*

**DOI:** 10.3390/ijms262211028

**Published:** 2025-11-14

**Authors:** Elena Michailovna Lazareva, Eugeniy Pavlovich Kazakov, Tatiyana Anatolievna Dilovarova, Neonila Vasilievna Kononenko, Larisa Ivanovna Fedoreyeva

**Affiliations:** 1All-Russia Research Institute of Agricultural Biotechnology, Timiryazevskaya 42, 127550 Moscow, Russia; 2Biological Department, M.V. Lomonosov Moscow State University, Leninskie Gory 1, 119991 Moscow, Russia

**Keywords:** metabolism, autophagy, *Nicotiana tabacum*, peptide, ultrastucture

## Abstract

The short peptide AlaGluAspLeu (AEDL) stimulates shoot and root development in *Nicotiana tabacum*. Growing tobacco in the presence of AEDL was found to induce autophagy and programmed cell death, as demonstrated using immunodetection of the autophagy marker ATG8 and cytochrome c in the cytoplasm, as well as the detection of DNA breaks using the TUNEL assay. A detailed study of the ultrastructure of *Nicotiana tabacum* root cells grown in the presence of AEDL using transmission electron microscopy revealed fundamental structural differences from control cells. Control cells contained only lytic vacuoles, while in the presence of AEDL, tobacco root meristem cells contained predominantly protein-storing vacuoles and amyloplasts with numerous starch granules in the stroma. Characteristic types of phagophores were identified, forming numerous small autophagosomes with cytoplasmic regions, multivesicular bodies, or concentric membranes, possibly with cytoskeletal elements. Expression of autophagy protein genes revealed a decrease in *TOR* expression, which promoted autophagy activation and prevented ATG13 phosphorylation. ATG8 gene expression significantly increased in the presence of the AEDL peptide. Schematic diagrams of autophagy processes in root cells of control plants and those grown in the presence of AEDL are presented. Based on these data, it was concluded that stimulation of tobacco plant development in the presence of the AEDL peptide at a concentration of 10^−7^ M occurs due to the activation of metabolic processes and autophagy. Moreover, the synthesis of metabolites exceeds the required amount of nutrients, which accumulate in vacuoles and leucoplasts.

## 1. Introduction

Cellular homeostasis requires strict regulation and coordination of the functions of various organelles [1]. To achieve this, plants have developed complex mechanisms for recycling intracellular components essential for life. These mechanisms allow plants to efficiently recycle nutrients, which is especially important under conditions of limited resource availability, and to properly utilize proteins, protein complexes, and even entire organelles that become damaged. For normal development, plants must maintain a balance between carbon uptake, storage, and growth [2,3]. Nutrient deficiency is also a stressful condition for plants. Carbon and nitrogen are essential nutrients for cells, from which plants synthesize carbohydrates such as sucrose and glucose, which are the main source of energy and structural elements, and, together with ammonium, participate in the process of amino acid biosynthesis. Sufficient amounts of carbon and nitrogen are crucial for plant growth and development [4,5]. Autophagy is one of the mechanisms that enables the breakdown of carbon sources to generate energy during carbon starvation. The degradation of damaged proteins and organelles releases amino acids, fatty acids, and other nutrients that the cell can use to synthesize new proteins, repair damaged structures, and maintain energy balance. Thus, autophagy ensures the remobilization of nutrients.

Autophagy is a highly regulated mechanism that ensures cell survival by recycling cellular components. It involves the formation of autophagosomes—double-membrane vesicles that capture cytosolic components, including damaged organelles (mitochondria or endoplasmic reticulum) and damaged proteins. These autophagosomes then fuse with lytic vacuoles, where their contents are destroyed, producing metabolites used by the cell to maintain its vital functions. Thus, in plant cells, vacuoles play a key role in maintaining cellular homeostasis, including the intracellular breakdown of various substances, maintaining turgor, and accumulating nutrients, ions, and secondary metabolites. Two functionally distinct vacuoles can exist within a single cell: the lytic vacuole (LV) and the protein storage vacuole (PSV) [6,7]. LVs contain hydrolytic enzymes for the degradation of damaged and waste material, while PSVs accumulate a large number of various proteins. The formation of autophagosomes begins with the formation of phagophores from the endoplasmic reticulum in the form of double-membrane flat sacs or tubes that surround areas of the cytoplasm [8].

Numerous proteins and receptors are involved in the activation of autophagy at different stages [1,9]. A subgroup of genes, called AuTophaGy-related genes (ATG genes), encode the corresponding ATG proteins. These proteins are involved in autophagosome biogenesis and autophagy control. More than 40 ATG proteins with various functions have been identified in eukaryotes [10,11]. It has been shown that several ATG proteins with different functions are simultaneously involved in the autophagy process. Proteins involved in the autophagy process have been grouped into four main functional groups [12,13]. The ATG1/ATG13 kinase complex initiates autophagosome formation, the class III phosphatidylinositol (PI) 3-kinase complex mediates vesicle formation, ATG8/ATG12 ubiquitin promotes phagophore expansion and maturation, and ATG9 can promote phagophore expansion by translocating membrane components from different sources [14]. ATG5 covalently binds to ATG12 to form an adduct, which then interacts with ATG16 to form the ATG5-ATG12/ATG16 complex. This complex transfers PE to ATG8 [15] to form the lipidated form ATG8-PE [16,17], which localizes to the autophagosome membrane and is commonly used as an autophagosome marker. Most ATG genes have been identified in *Arabidopsis thaliana* based on sequence similarity to yeast genes and have been shown to be required for autophagy [18,19].

Peptides are widespread in plants. Interest in their study has increased in recent years due to the discovery that peptides actively participate in the regulation of many physiological processes, leading them to be called peptide hormones [20]. Like phytohormones, peptides are involved in many cellular processes, intercellular communication, and signal transduction. The intercellular communication system based on peptide–receptor interactions plays a crucial role in plant development and function. Peptide hormones are recognized by their receptors, which transmit signals to subsequent targets and interact with multiple pathways to fine-tune plant growth.

We previously found that the tetrapeptide AlaGluAspLeu (AEDL) at a concentration of 10^−7^ M in the medium stimulates the development of the root system of tobacco *Nicotiana tabacum* [21]. It was suggested that AEDL is involved in the regulation of cell proliferation and differentiation, similar to the CLE40 peptide [22]. Rapid growth and more intensive development of the plant require additional nutrition. Based on this, the aim of our study was to investigate the structural features of root meristem cells and the molecular processes that lead to accelerated plant development.

## 2. Results

### 2.1. Morphometric Parameters

As shown in the data presented (Figure 1 and Table 1), the AEDL peptide activates the development of *Nicotiana tabacum*. Tobacco grown in the presence of 10^−7^ M AEDL exhibits an increase in shoots and root system size compared to the control sample. The total wet and dry weights of the tobacco exceed those of the control sample (by 1.7 and 1.4 times, respectively). The increase in total weight of tobacco grown in the presence of the AEDL peptide is due to both shoot growth and a robust root system.

### 2.2. Immunodetection

It is known that autophagy proteins ATG are conserved across yeast, animal, and plant cells. In our study, we detected the ATG8 protein in normal root cells of 28-day-old *Nicotiana tabacum* seedlings and 28-day-old seedlings grown in the presence of the AEDL peptide.

One way to provide nutrients to actively developing plants is through the activation of autophagy. Therefore, we performed immunodetection of the ATG8 protein, an autophagy marker, on the membranes of autophagosomes in macerated root meristem cells of *Nicotiana tabacum.*

In the cytoplasm of outer and inner cortex meristem cells of control, ATG8 marks the surface of small autophagosomes around the nucleus and the nucleolus region.

The cytoplasm of outer cortical cells of tobacco grown in the presence of the AEDL peptide contains larger vacuoles than in the control (phase) (Figure 2 and Figure 3). ATG8 marks the nuclear envelope, the surface of vacuoles near the nucleus, and the nucleolar region. In some outer cortex cells, ATG8 is localized both in the cytoplasm and as distinct bright spots on the surface of autophagosomes (white arrows).

### 2.3. Detection of DNA Breaks into Meristem Nuclei Using the TUNEL Method

The formation of DNA breaks and the release of cytochrome c from mitochondria are reliable markers of cell death accompanying plant development (Table 2). It was found that in *Nicotiana tabacum* root cells grown in the presence of the short AEDL peptide, the number of cells with DNA breaks increased (by 1.6 times) compared to tobacco root cells grown under control conditions.

### 2.4. Immunodetection of Cytochrome C into Meristem Cells

Cytochrome c was immunodetected in cytoplasm of meristem root cells *Nicotiana tabacum*. The presence of the AEDL peptide leads to an increase in the permeability of the outer mitochondrial membrane and the release of cytochrome c into the cytoplasm. The number of cells with cytochrome c in cytoplasm was about 2.6 times higher compared to tobacco control root cells (Table 3).

In the presence of AEDL, an increase in the number of dying cells with the release of cytochrome c into the cytoplasm was detected in tobacco root cells, which indicates a mitochondrial pathway of their death.

### 2.5. TEM Analysis

In the dense cytoplasm of the endoderm cells of the apical meristem of the control *Nicotiana tabacum* plants, round nuclei with chromosomes and a nucleolus, numerous small oval vacuoles forming chains near the nuclei, autophagosomes, mitochondria, and large plastids with a dark stroma were visible. The cells of the outer root cortex had an elongated shape. Large vacuoles and autophagosomes were visible in their cytoplasm (Figure 4a–c). Single large autophagosomes with concentric membranes or cytoskeletal elements were present in some cells (Figure 4c). In some cells of the endoderm meristem, double-membrane sickle-shaped phagophores with vacuoles inside were present in the cytoplasm near the cell wall (Figure 4d). In many cells of the inner cortex, exocytosis of the contents of autophagosomes into the cell wall observed (Figure 4f).

In the presence of the peptide, the morphology of tobacco root cells changes. It is noticeable that the vacuoles in the endoderma meristem cells are larger than in the cells of the control plants. These are vacuoles for storing various proteins. In addition to the storage vacuoles, autophagosomes are observed. Plastids are present in the cytoplasm. Unlike the endoderma meristem cells, the cortex cells contain vacuoles with membrane invaginations and numerous plastids with starch grains in the stroma, typical amyloplasts in appearance (Figure 5Right). Typical autophagosomes are also detected.

In plants grown in the presence of the AEDL peptide (Figure 5Left,Right), the morphology of the root cells differs from the root cells of the control plants (Figure 4a,b). In the cytoplasm of the meristem endoderma cells, only large vacuoles storing various proteins and autophagosomes are present (Figure 5a–f). Amyloplasts with numerous starch grains in the stroma are detected in the outer cortex cells.

In the presence of AEDL, various phagophores formed in the cytoplasm of tobacco root cells. Along with the typical crescent-shaped phagophores (Figure 6a,b), phagophores appear in the form of thin tubes or thin-walled sacs. On the periphery of the cells, areas of cytoplasm are visible inside small autophagosomes (Figure 6a–g).

In contrast to the control, in the presence of AEDL, various autophagosomes with numerous multivesicular bodes (MVBs) are formed in the cytoplasm of tobacco root cells (Figure 7d,e).

In the presence of AEDL in the cytoplasm of the cells of the tobacco root outer cortex, starch grains begin to accumulate in the plastids, forming typical amyloplasts (Figure 8).

### 2.6. Genes Expression

In the roots of *Nicotiana tabacum,* grown in the presence of the AEDL peptide, the expression of the *TOR* gene is lower (1.1 times) than in the control (Figure 9). At the same time, the expression level of *ATG1a*, especially *ATG13c*, is higher than in the control sample (1.1 times and 1.4 times, respectively). The expression level of *ATG9* in the peptide is slightly higher than in the control (1.1 times). While the expression level of *ATG5* is practically independent of the presence of the AEDL peptide in the nutrient medium, the expression activity of the *ATG12* and *ATG16* genes increases, especially *ATG16* (1.1 times and 1.2 times, respectively). It is interesting to note that the basal expression level of *ATG4* in the control exceeds that in the roots of tobacco grown in the presence of the AEDL peptide (1.4 times). At the same time, the activity of *ATG8c* gene expression in the roots of *Nicotiana tabacum* in the presence of AEDL significantly exceeds that of the control (more than 1.7 times).

## 3. Discussion

Plant metabolism includes photosynthesis and respiration, as well as the synthesis and breakdown of organic compounds in plant cells. Primary metabolism of carbohydrates, amino acids, lipids, and nucleic acids includes all chemical reactions that produce energy for cellular processes and supply building blocks, as well as ensure basic cellular functions and, consequently, normal plant growth and development [11]. Autophagy is induced by nutrient levels in plants [23]. Impaired autophagy leads to reduced plant growth due to the inability to adapt to changes in metabolic status during nutrient deficiency [1,15,16,17,18,19].

Research on autophagy and metabolism is conducted on primary metabolites. Therefore, these two processes are interconnected. There are two main types of vacuoles in plant cells: lytic vacuoles (LVs) and protein storage vacuoles (PSVs) [24]. A study on the remodeling of vacuoles in root meristem cells at the early stages of tobacco seed germination was conducted by H. Zheng and L. A. Staehelin [8]. Large amounts of storage proteins accumulate in the seeds of higher plants. Studies have shown that one of the sites of protein reserve mobilization may be radicle meristem cells, where vacuole remodeling can occur [25]. Meristematic cells of the root tip contain numerous small vacuoles, which fuse in differentiated cells to form larger vacuoles. Interest in studying vacuole transformations has increased due to increased research on metabolic processes in storage and lytic vacuoles, as well as the need to understand vacuolar transport and metabolic systems [26]. Storage proteins can be deposited in protein bodies located in the cytoplasm. These protein bodies can subsequently be transported to protein storage vacuoles. Refs. [25,27,28] used TEM to demonstrate the presence of protein storage vacuoles in root meristem cells, the formation of which resembles the initial stages of autophagy. Protein body degradation occurs via autophagosomes. The time interval between autophagy and autolysis, which is typically short in animals and yeast, significantly increased in protein storage vacuoles in plants. Olbrich et al. [25] demonstrated that a protein storage vacuole can become autophagic.

In the cells of the epidermis and outer cortex of the root meristem, large vacuoles are formed from which storage proteins are released [8]. In the cells of the inner cortex (endoderm) of tobacco grown in the presence of AEDL, we identified numerous PSVs and autophagosomes (Figure 5Left), most likely formed after phagophores surrounded cytoplasmic materials for further degradation. Paris et al. [29] showed that large LVs of vegetative cells develop as a result of the fusion of individual PSVs and LVs. We observed variants of the convergence of protein storage vacuoles, which will later lead to the formation of a large vacuole, in cortical cells in the presence of the AEDL peptide.

It is hypothesized that the short peptide AEDL acts similarly to peptide hormones in plant cells, promoting intensive plant development [21]. A significant increase in the root system and the size of the above-ground portion of *Nicotiana tabacum* grown in the presence of the AEDL peptide (Figure 1) was demonstrated. Maintaining a high metabolic rate and the necessary additional nutrients may result from the activation of autophagy.

The distribution of the autophagy marker protein Atg8 was studied in cells of the inner and outer cortex of the tobacco root tip meristem. We found that Atg8 marked the membrane surface of autophagosomes of various sizes and shapes near the nuclei, as well as the surface of the nuclear envelope, in both inner cortex cells of control plants and plants grown in the presence of AEDL (Figure 2A and Figure 3B). Autophagosomes in the cells of the outer root cortex in the presence of the peptide were larger than in the control and did not have a rounded shape; the Atg8 protein in the form of individual grains marked individual sections of their membranes (Figure 3B).

The arrangement of vacuoles, autophagosomes, and phagophores revealed by an ultrastructure study of root meristem cells in control *Nicotiana tabacum* plants (Figure 4b–d) corresponds to the immunolocalization patterns of the Atg8 protein on the surface of these structures (Figure 2A,B). In the cells of the outer cortex, in the presence of AEDL, Atg8 was observed both in the cytoplasm, indicating the intensive translation of Atg8 on ribosomes, and as separate bright spots, apparently in invaginations of the tonoplast of autophagosomes (Figure 3B). In endoderm cells, Atg8 is localized on the surfaces of autophagosomes and the nuclear membrane (Figure 3A).

We showed that the number of cells with DNA breaks and cytochrome c release from mitochondria increased in the root meristem of plants grown in the presence of AEDL (Table 2 and Table 3). A study of the ultrastructure of the cells of the tobacco root meristem cortex revealed differences in the morphology of organelles in plant cells grown in the presence of the AEDL peptide and in the control.

Autophagosomes were detected in the meristem cells of tobacco roots grown in the presence of AEDL using immunodetection of the autophagy marker Atg8 and transmission electron microscopy. Ultrastructural examination revealed that both large and small vacuoles of the meristem cells were storage vacuoles for various proteins (Figure 5Left,Right). These vacuoles formed chains and could fuse (Figure 5e). Interestingly, the vacuoles of the outer cortex cells of the root meristem differed in structure from the vacuoles of the endoderm cells. The vacuoles of the endoderm had an irregular shape and tonoplasts with numerous invaginations (Figure 5c,e). We believe that the surfaces of such vacuoles are ready for interactions with targets (proteins), or vacuoles and autophagosomes.

Phagophores of various shapes were found in the cells of the endodermis and outer cortex of the root meristem (Figure 6). In the endodermis cells, crescent-shaped phagophores were found surrounding areas of cytoplasm on both sides (Figure 6a). Sometimes the surface of these phagophores was wavy (Figure 6b). In the cytoplasmic cortex of the cortex cells, phagophores in the form of thin tubes or thin-walled sacs (since we see these tubes in ultrathin sections) were found near the cell walls (Figure 6c). Sometimes, lipid droplets are located near such phagophores. We cannot exclude the possibility that they are an additional source of membrane lipids. Similar structures in plant cells were observed by van Doorn and et al. [30]. Some crescent-shaped phagophores surrounded areas of cytoplasm with ribosomes, forming very small autophagosomes (Figure 6d–f,h). No similar autophagosomes were detected in the root cells of control plants. Similar structures, termed cytoplasmic prelytic vacuolar domains, were described in the cells of 3–6-day-old tobacco seedlings.

In tobacco root cells, in the presence of AEDL, we detected different types of autophagosomes (Figure 7). Autophagosomes containing numerous double-membrane vesicles were frequent (Figure 7d,e). These vesicles were exocytosed into the apoplast (Figure 7b, blue star). Multivesicular bodies (MVBs) surrounded by a membrane with invaginations were detected in the autophagosome (Figure 7d). Autophagosomes with concentric membranes or cytoskeletal elements were detected (Figure 7e). Extended ER cisternae were located near such autophagosomes. Many autophagosomes had a complex structure after capturing different areas of the cytoplasm (Figure 7a,b), which indicates their non-selective function. Such autophagosomes can form in cells with intensive metabolism.

In the leucoplasts of endoderm cells of the tobacco root meristem in the control, individual starch grains and plastoglobuli were observed (Figure 8c). In contrast, in the outer root cortex cells grown in the presence of the AEDL peptide, leucoplasts accumulated numerous starch grains, the glucose from which was likely produced in leaf chloroplasts and then transported to root cells. This morphology is characteristic of typical amyloplasts (Figure 8a,b,d,e). These amyloplasts of the outer root cortex are most likely involved in processes of enhanced carbohydrate metabolism.

Thus, intensive proliferation of *Nicotiana tabacum* cells grown in the presence of AEDL accelerates plant development and is facilitated by the accumulation of starch in amyloplasts and proteins in PSVs, the detection of which indicates enhanced metabolic processes compared to control plant cells. The production of additional nutrients is ensured by the activation of autophagy, which digests proteins, carbohydrates, and other substances in vacuoles, and by an increase in the number of cells with DNA breaks in the nuclei and the release of cytochrome c into the cytoplasm, leading to PCD.

In *Nicotiana tabacum* root meristem cells grown in the presence of the AEDL peptide, autophagy is induced by increased metabolism. This hypothesis is based on the fact that *Nicotiana tabacum* grows more rapidly in the presence of AEDL and, consequently, consumes more nutrients (Figure 1). A study of the ultrastructure of tobacco root cells revealed the accumulation of starch in leucoplasts and proteins in the PSV. Activation of autophagy, which involves the digestion of proteins, carbohydrates, and other substances in vacuoles, may provide additional nutrition to seedlings. Currently, the most studied process is the process of initiation of autophagosome formation.

There are both TOR-dependent and -independent pathways of autophagy regulation [31]. Rapamycin (TOR-serine/threonine protein kinase) is considered to be one of the central negative regulators of the autophagy initiation process, controlling autophagy mediated by the ATG1/ATG13 complex [32]. *TOR* activity is reduced to regulate autophagy in response to nutrient concentration [33]. We noted a decrease in *TOR* expression activity in *Nicotiana tabacum* roots in the presence of peptide AEDL compared to the control, although not significant (Figure 1). The target of rapamycin is phosphatidylinositol 3-kinase-related kinase (PI3K), which functions as a Ser/Thr protein kinase and is a negative regulator of autophagy [33,34,35]. PI3K is involved in the phosphorylation ATG. It also regulates growth and protein synthesis depending on the availability of nutrients and growth factors [34,35,36]. In yeast, TOR negatively regulates autophagy by activating the phosphorylation reaction of ATG13, thereby decreasing its affinity for ATG1 under nutrient-rich conditions and reducing autophagic activity. During starvation, TOR is inactivated, preventing ATG13 phosphorylation and increasing the binding between ATG13 and ATG1. ATG1 is also involved in the activation of autophagy-specific PI3-kinase [37]. The PI3K complex mediates the production of PI3P at the PAS and further recruits PI3P effectors such as ATG18 [38] and the ATG12-ATG5-ATG16 complex [39]. Localized enrichment of PI3P at the PAS is a hallmark of autophagy, which is observed in mammals, yeast, and plants.

The expression level of the *ATG1a* gene in tobacco roots grown in the presence of AEDL is slightly higher than in the control tobacco (Figure 9). However, the level of ATG13c in *Nicotiana tabacum* in the presence of the peptide increases by 1.4 times compared to the control. There is evidence that ATG1 kinase is not required for the activation of autophagy under long-term carbon deficiency, although it is required for the activation of autophagy under short-term carbon deficiency, as well as under nitrogen deficiency [40]. During starvation, the first step of phagophore formation requires the assembly of the ATG1/ATG13 complex to activate autophagy. The main role of the ATG1 complex is to recruit downstream regulators for the assembly of PAS (preautophagosomal structure) and in membrane/vesicle bundling for phagophore development (including ATG9 vesicles) [41]. ATG1a promotes the activation of PI3P, as well as the activation of the ATG12-ATG5-ATG16 complex [39]. We also suggest that ATG1a may participate in the lipidation process together with ATG9 and, thus, actively participate in the initiation of autophagophore formation. A more pronounced activation of the *ATG13c* gene compared to *ATG1a* may indicate that ATG13c is involved in processes other than recruiting downstream regulators for PAS assembly together with ATG1a. The transmembrane protein ATG9 in complex with PE is involved in the formation of the membrane of the preautophagosomal structure and promotes the expansion of the isolation membrane. In addition, ATG9 is believed to regulate the formation of phagophores from the endoplasmic reticulum of plants [42]. It has been shown that ATG9 may also be involved in other processes. In yeast, it was found that ATG9 vesicles can be used as substrates for the PI3K complex to produce PI3P, which recruits effectors and promotes lipidation of ATG8 [43]. Despite slight activation of the *ATG9* gene in tobacco roots in the presence of the AEDL peptide, there is no increase in the size of autophagosomes, which is compensated by the formation of a larger number of small autophagosomes at the periphery of cells, in contrast to large autophagosomes in control cells.

The level of *ATG4* expression in tobacco roots grown in the presence of the AEDL peptide is lower than in the control, which may indicate that ATG4 is involved only in the activation of ATG8, and the main processing of denatured proteins occurs in the vacuoles of the cytoplasm, the size of which is significantly larger than in control cells.

It can be hypothesized that in the presence of the AEDL peptide, numerous denatured cytosolic proteins in root meristem cells are more actively transported and accumulated in vacuoles for further processing, simultaneously forming small autophagosomes that also supply the cells with nutrients. This is confirmed by an increase in ATG8 expression in tobacco in the presence of the AEDL peptide, significantly exceeding that in the control.

TOR is an important negative regulator of autophagy. Under nutrient supply, *TOR* is active and prevents the assembly of the ATG1/ATG13 kinase complex by activating ATG13 phosphorylation. Under nutrient deficiency, *TOR* is inactivated and promotes ATG13 dephosphorylation and ATG1/ATG13 complex formation [31,32,33]. Figure 10B shows that *TOR* activity was suppressed in tobacco roots grown in the presence of AEDL, which was accompanied by the formation of the ATG1/ATG13 complex. As shown by transmission electron microscopy, tobacco root cells in the presence of the AEDL peptide (in contrast to control cells) form multiple starch grains (bodies), indicating a long-term deficiency of fixed carbon. As shown in *Arabidopsis*, ATG1 kinase activation does not occur under these conditions [37]. We also noted that *ATG1a* is not activated in the presence of the AEDL peptide, in contrast to the significant activation of *ATG13c*.

There is evidence that nitrogen starvation leads to impaired autophagy in Arabidopsis and a decrease in lipid levels [18]. Probably, in the presence of AEDL in tobacco root cells, a lipid deficiency is observed, which is manifested in the formation of small autophagosomes, although in greater quantities than in control tobacco root cells. Activation of the autophagy process in the presence of AEDL is not accompanied by activation of ATG9, which is responsible for the formation and expansion of phagophores. In addition, according to TEM analysis, small autophagosomes are formed in tobacco cortex cells in the presence of AEDL, which confirms the assumption of lipid deficiency.

Another feature of autophagy in tobacco cells in the presence of AEDL is the activation of the key autophagy gene ATG8. As was found in Arabidopsis, almost all ATG8 isoforms are intensely expressed under nitrogen starvation [38,39]. Under carbohydrate starvation, only a small number of ATG8 isoforms are activated. According to our data, the AEDL peptide promotes a significant increase in the expression level of only ATG8c, which indicates that under these conditions there is a deficit of fixed carbohydrate, the deficiency of which is compensated by the transport of glucose from the leaves and its accumulation in starch grains in plastids.

It should be emphasized that the ultrastructural analysis was conducted on cortical and meristem cells of tobacco seedlings located at the root tip, near the stem cell zone. While the AEDL peptide had previously been shown to penetrate root cells, it was not detected in the meristem zone. It was hypothesized that the AEDL peptide is involved in regulating the proliferation–differentiation process, similar to the CLE40 peptide. This hypothesis suggests that the meristem cells bordering the initial cells (stem cells) accumulate nutrients (proteins and carbohydrates) necessary for cell proliferation.

## 4. Materials and Methods

### 4.1. Plant Material

Tobacco seeds (*Nicotiana tabacum* L.) of the Samsun variety were placed in test tubes containing Murashige-Skoog (MS) medium without hormones (control samples); the medium of test samples was supplemented with 10^−7^ M AEDL. Cultivation was carried out in a light room at a temperature of 26 °C; the photoperiod duration was 16 h. The experiment was carried out for 4 weeks in 3 replicates. After 28 days, tobacco plants grown in different conditions were selected and morphometric parameters were determined. Roots and shoots were used for further studies. The AlaGluAspLeu peptide was synthesized by IQChem (St. Petersburg, Russia).

### 4.2. Light Microscopy

Root tips (5 mm) were fixed in 4% paraformaldehyde solution (Sigma Aldrich, St. Louis, MI, USA) in PHEM buffer, pH = 6.9 (60 mM PIPES (Sigma Aldrich, USA)), 25 mM HEPES (Sigma Aldrich, USA), 10 mM EGTA (Sigma Aldrich, USA), and 2 mM MgCl_2_ (Sigma Aldrich, USA) for 1.5 h at room temperature. The fixative was washed with PHEM buffer. Samples were incubated for 7 min in 0.4 M mannitol containing 4% cellulase (Sigma Aldrich, USA) and 5 mM EGTA and then washed with PBS buffer; after incubation in the enzyme, the roots were transferred to coverslips and separated into individual cells. The prepared preparations were dried in a refrigerator at a temperature of +4 °C for 24 h.

### 4.3. TUNEL Assay

To detect DNA strand breaks using the dUTP terminal deoxynucleotidyl transferase nick end labeling (TUNEL) assay, a TUNEL assay kit (CAS number: K3000, Sileks, Moscow, Russia) was used. Macerated cell preparations were permeabilized in 0.5% Triton X100 (Sigma Aldrich, USA) in PBS for 30 min; then, after washing twice with buffer, they were placed in cocodelate buffer (pH 7.4) containing 20 U/μL terminal deoxynucleotidyl transferase (Sileks, Moscow, Russia), 3′-labeled probes with 10 mM dATP (Sileks, Moscow, Russia) and 1 mM fluorescein (Sileks, Moscow, Russia). The reaction was stopped by placing the preparations in 2× SSC solution for 15 min. After washing twice with buffer, the preparations were embedded in Mowiol U-44 (Hoechst, Germany) with the addition of DAPI (1 µL/1 mL). Samples were analyzed under an Olympus BX51 fluorescence microscope (Tokyo, Japan), ×100 objective. Images were obtained using a Color View digital camera and Cell software (Germany).

### 4.4. Detection of Cytochrome C

Before immunocytochemical detection of mitochondria, the preparations were placed in PBS for 5 min and transferred to a solution of 0.5% Triton X-100 in PBS for 30 min. Then they were washed in PBS and incubated for 16–18 h at room temperature with rabbit polyclonal antibodies (diluted 1:100 in PBS + 0.1% BSA) against cytochrome c (AS08 343A, Agrisera, Vännäs, Sweden). The preparations were then washed and incubated for 45 min at 37 °C with goat antibodies (1:25 dilution in PBS + 0.1% BSA) against rabbit IgG conjugated with Texas Red fluorochrome (Sigma Aldrich, USA), used as secondary antibodies, washed, stained with DAPI, and mounted in Mowiol U-88 (Hoechst, Germany). Samples were analyzed under an Olympus BX51 fluorescence microscope (Tokyo, Japan), ×100 objective. Images were obtained using a Color View digital camera and Cell software (Germany).

### 4.5. Immunodetection of ATG8 Antibodies

The preparations were placed in PBS for 5 min and transferred to a solution of 0.5% Triton X-100 in PBS for 30 min. Then they were washed in PBS and incubated for 16–18 h at +4 °C temperature with mouse monoclonal antibodies LC3B Ab-AF4650 (diluted 1:100 in PBS + 0.1% BSA). The preparations were then washed and incubated for 45 min at 37 °C with antibodies (1:50 dilution in PBS + 0.1% BSA) against rabbit IgG conjugated with Texas Red fluorochrome (Sigma Aldrich, USA), used as secondary antibodies, washed, stained with DAPI, and mounted in Mowiol U-88 (Hoechst, Germany). Samples were analyzed under an Olympus BX51 fluorescence microscope (Japan), ×100 objective. Images were obtained using a Color View digital camera and Cell software (Germany).

### 4.6. TEM Analysis

Root tips (4–5 mm) were fixed for 24 h in 2.5% glutaraldehyde (Merck, Darmstadt, Germany) prepared on 0.1 M Sorensen’s phosphate buffer (pH 7.2) and containing 1.5% sucrose. Then samples were washed, post-fixed in 1% OsO4 (Sigma-Aldrich, St. Louis, MI, USA), dehydrated in ethanol of increasing concentrations (30°, 50°, 70°, 96°, and absolute acetone), and embedded in a mixture of Epon-812 (Merck, Darmstadt, Germany), according to the standard protocol.

For electron microscopy, samples were sectioned with a diamond knife using ultrathin sections stained with uranyl acetate and lead citrate, and then analyzed and photographed using an electron microscope (Jeol-1400, Japan).

### 4.7. Total RNA Isolation and Gene Expression Analysis

The analysis was performed using a standard RNA isolation kit “Extran RNA Syntol” (Moscow, Russia). RNA was isolated from the roots and shoots of *N. tabacum* grown under different conditions. cDNA was synthesized by reverse transcription using standard methods (Syntol, Moscow, Russia). The concentration of cDNA was determined spectrophotometrically using an IMPLEN nanophotometer. RT-PCR using SYBR Green I (Syntol) was performed on a CFX 96 real-time thermal cycler (BioRad, Hercules, CA, USA). Information on the structure of the *ATG* genes in N. tabacum was obtained from NCBI. Primers for the genes were synthesized by Syntol. The *GaPDh* gene was used as a reference gene. Each RT-PCR reaction was performed in triplicate.

### 4.8. Statistical Analysis

Statistical processing of experimental data was carried out using one-way analysis of variance (ANOVA) and Student’s *t*-test (R version 4.3.1) with significant differences at *p* < 0.05. The least significant difference method was used to test significance. Values are presented as means ± standard deviations of triplicate biological replicates.

## 5. Conclusions

It was shown that *Nicotiana tabacum* grown in the presence of the AEDL peptide had larger size and bigger biomass. According to our observations, the increase in plant size is associated with an increase in the intensity of cell metabolism processes. For the first time, the ultrastructure of *Nicotiana tabacum* root cells grown in the presence of the short AEDL peptide was studied, which made it possible to identify their characteristic features and fundamental differences from the cells of control plants.

Firstly, in the cells of the meristem and root cortex in the presence of AEDL, only protein storage vacuoles were detected, while in the control plants, various lytic vacuoles were present. Secondly, in the presence of AEDL, the leucoplasts (plastids) of the root were transformed into amyloplasts due to the accumulation of starch in the stroma, most likely coming from the cells of the above-ground parts of the plant. On the other hand, in the cells of control plants, leucoplasts had a dense stroma with single thylakoids. Changes in the structure of these organelles confirm our assumption about the intensification of protein and carbohydrate metabolism, which ensures an increase in plant size in the presence of AEDL. An additional contribution to this process is made by the formation of small specific autophagosomes.

Thirdly, characteristic types of phagophores that form autophagosomes were identified. Fourthly, small autophagosomes with characteristic contents were detected in root cells in the presence of AEDL: these were cytoplasmic regions with ribosomes or with multivesicular bodies or with concentric membranes or cytoskeletal elements.

Expression of autophagy protein genes revealed a slight inhibition of *TOR* expression, preventing phosphorylation of ATG13, which is typical for a lack of nutrients in cells. It was noted that the expression of ATG8 genes is significantly activated in the presence of the AEDL peptide. Based on the study of the expression of autophagy protein genes, a scheme of autophagy was proposed, occurring both in control tobacco cells and in the presence of the AEDL peptide, characteristic of processes with a lack of carbohydrates.

Thus, for the first time it was shown that in the presence of the AEDL peptide at a concentration of 10^−7^ M, intensification of metabolic processes occurs, mainly protein and carbohydrate. The formation of small autophagosomes contributes to more intensive digestion of the necessary building elements for more intensive growth of tobacco seedlings.

## Figures and Tables

**Figure 1 ijms-26-11028-f001:**
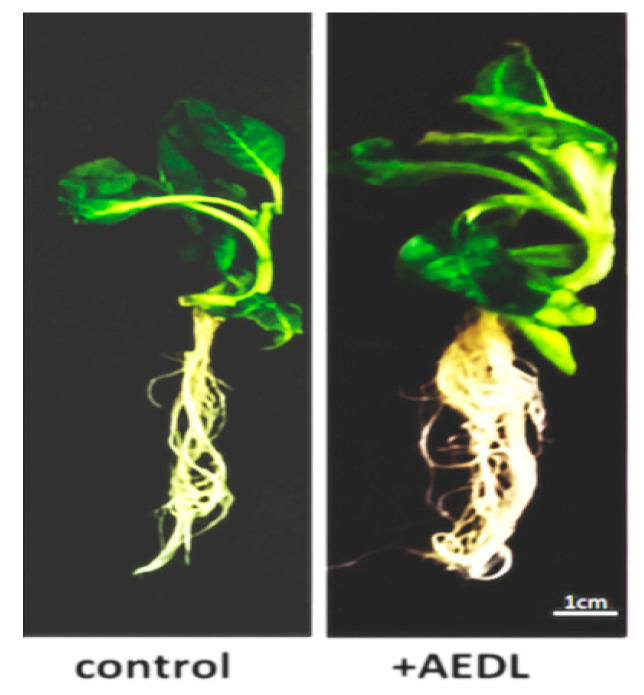
*Nicotiana tabacum*, grown in different conditions.

**Figure 2 ijms-26-11028-f002:**
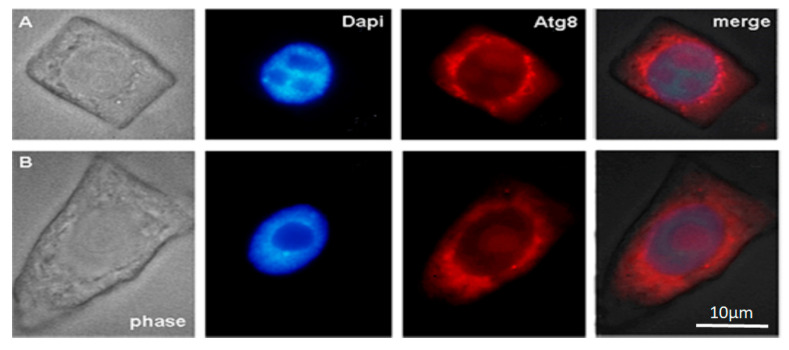
Immunodetection protein ATG8 into control macerated cells of the apical meristem tobacco root tips. (**A**) Meristem root cells of inner (endoderma) cortex and (**B**) outer cortex. Scale bar, 10 µm.

**Figure 3 ijms-26-11028-f003:**
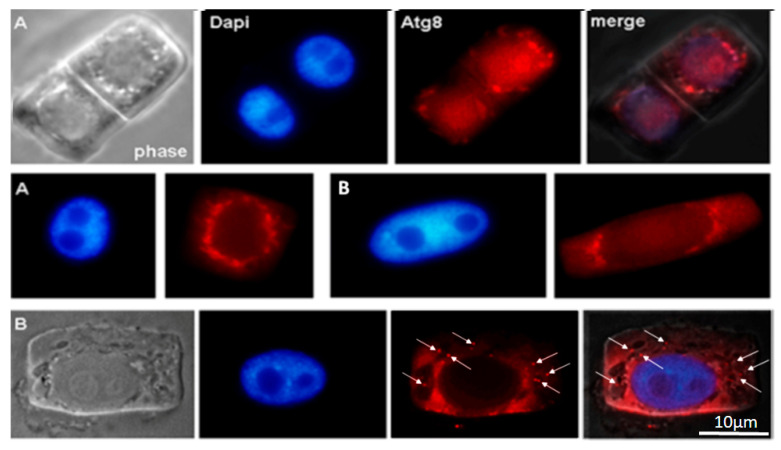
Immunodetection protein ATG8 into macerated cells of the apical meristem tobacco root tips, grown in the presence of the AEDL peptide. (**A**) Meristem root cells of inner (endoderma) cortex and (**B**) outer cortex. The distinct bright points on the surface of autophagosomes (white arrows). Scale bar, 10 µm.

**Figure 4 ijms-26-11028-f004:**
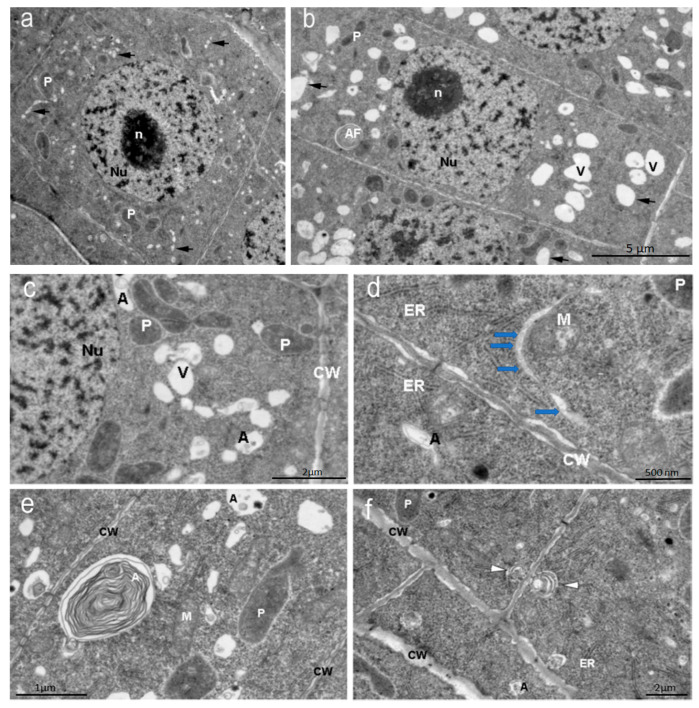
*Nicotiana tabacum* meristem root cells of inner (endoderma) cortex and outer cortex cells, grown in control condition. (**a**) Meristem inner (endoderma) cortex cell and (**b**) outer cortex cell. Bar ×2500. (**c**) Vacuoles, autophagosomes, and plastids of the meristem cell cytoplasm. Bar ×6000. (**d**) Double-membrane phagophores with small vesicles inside (arrows). Bar ×15,000. (**e**) Autophagosomes with concentric membranes or cytoskeletal elements or vesicles. Bar ×8000. (**f**) Autophagosomes during exocytosis (white arrows) into the cell wall. Bar ×4000. Nu—nucleus; n—nucleolus; P—plastids; CW—cell wall; A—autophagosomes; V—vacuoles.

**Figure 5 ijms-26-11028-f005:**
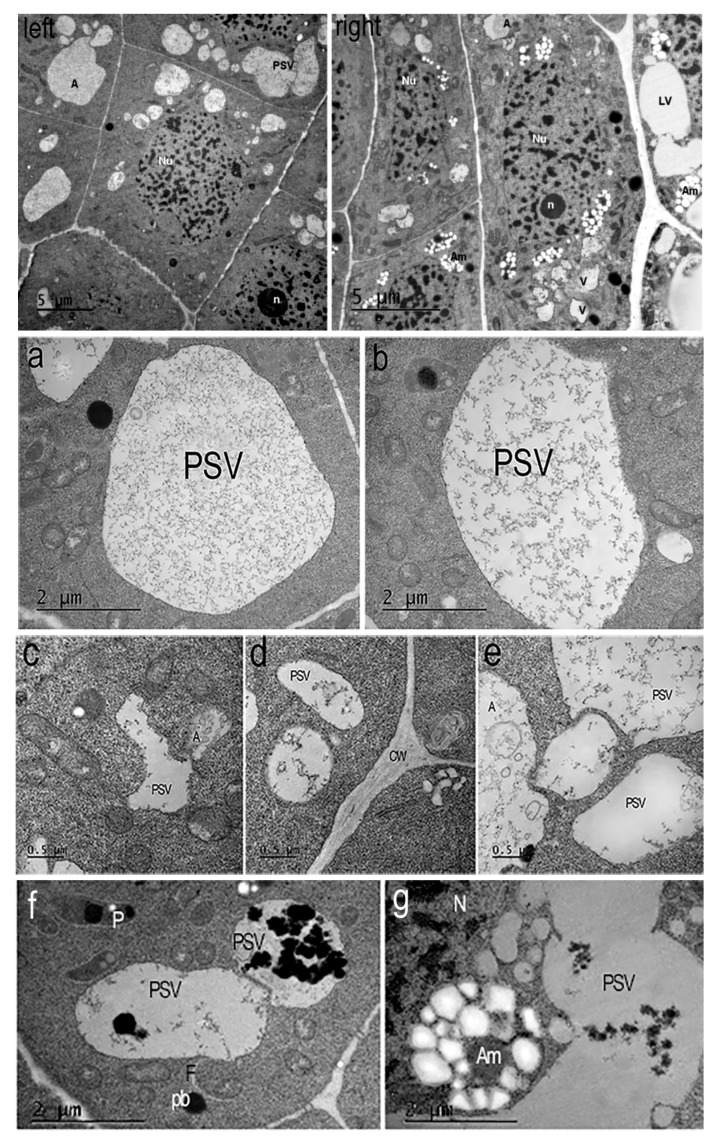
*Nicotiana tabacum* meristem root cells of inner (endoderma) cortex and outer cortex cells, grown in the presence of AEDL. (**Left**) The endoderma cell contains a round nucleus (Nu) with invaginations of the nuclear membrane, chromocenters, chromatin fibrils, and a nucleolus (n). The cytoplasm contains large round protein storage vacuoles (PSVs), small autophagosomes, lipid droplets, mitochondria, and plastids. Bar ×2500. (**Right**) The cortex cells contain oval nuclei (Nu) with a nucleolus (n), chromatin fibrils, and chromocenters. The cytoplasm contains numerous plastids/amyloplasts (Am) with starch grains and plastoglobuli, lytic vacuoles (LV), and autophagosomes (A). Bar ×2500. (**a**–**g**) Typical protein storage vacuoles (PSVs) in the cytoplasm of *Nicotiana tabacum* root cells in the presence of AEDL. (**a**,**b**,**e**) Large oval protein storage vacuoles of cells endoderma. A—autophagosome, PSV - protein storage vacuole. Bar ×12,000. (**e**) Bar ×25,000. (**c**) Small vacuoles with membrane invaginations in the outer cortex cell or (**d**) rounded tonoplasts in the endoderma cell. Bar ×50,000. (**f**,**g**) Degradation of protein into vacuoles and amyloplasts with starch of outer cortex cells. Am—amyloplast; pb—protein body; F—phagophore; PSV—protein storage vacuole; N—nucleus. Bar ×12,000.

**Figure 6 ijms-26-11028-f006:**
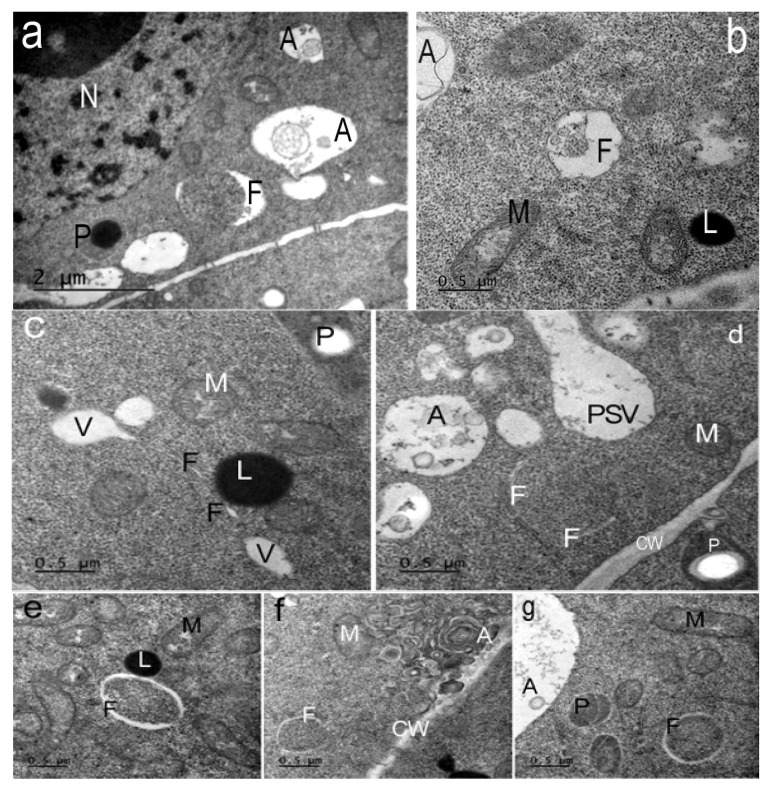
Types of phagophores in the cells of the *Nicotiana tabacum* root meristem, grown in the presence of the peptide AEDL. Phagophores (F) of various morphologies found in the root cells. (**a**) Phagophores (F) can have the shape of a large crescent. A—autophagosomes; P—plastoglobuli of plastids; n—nucleolus. Bar ×25,000. (**b**) Phagophores with membrane invaginations. A—autophagosome; M—mitochondria; L—lipid drop. (**c**) Phagophore (F) in the form of a thin tube extending from a vacuole. V—vacuoles; M—mitochondria; P—plastids; L—lipid drop. (**d**) Several thin tubes-phagophores (F) around a section of cytoplasm. A—autophagosome; CW—cell wall; P—plastids with a starch grain; PSV—protein storage vacuole. (**e**,**f**,**g**) Many phagophores form autophagosomes with sections of cytoplasm inside. (**e**) Phagophores (F); M—mitochondria; L—lipid drops are visible in the cytoplasm. (**f**) Phagophore (F); autophagosomes (A) with concentric membranes or cytoskeletal elements. CW—cell wall; M—mitochondria. (**g**) Phagophore (F); A—autophagosome; M—mitochondria; P—plastids. (**b**–**g**) Bar ×50,000.

**Figure 7 ijms-26-11028-f007:**
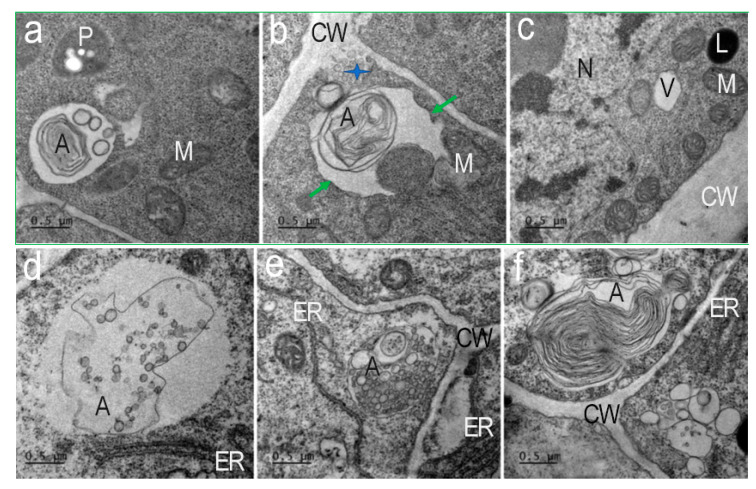
Types of autophagosomes of *Nicotiana tabacum* root cells in the presence of AEDL. Autophagosomes containing concentric membranes or cytoskeletal elements and double-membrane vesicles found near the cell wall of the cells (**a**,**b**,**f**). (**a**) Autophagosome with vesicles and concentric membranes. M—mitochondria; P—plastid. (**b**) Autophagosomes at the moment of absorption of a section of cytoplasm (green arrows) next to a multivesicular body extruded into the cell wall (CW, blue star). (**c**) Nuclei (Nu) have invaginations of the nuclear membrane; in the cytoplasm there are many mitochondria (M), individual lytic vacuoles (V), and lipid drop (L). (**d**–**f**) Autophagosomes (A) with multivesicular bodies, cisterns of the endoplasmic reticulum (ER), or cytoskeleton elements are nearly visible on the cell wall. CW—cell wall. Bar ×50,000.

**Figure 8 ijms-26-11028-f008:**
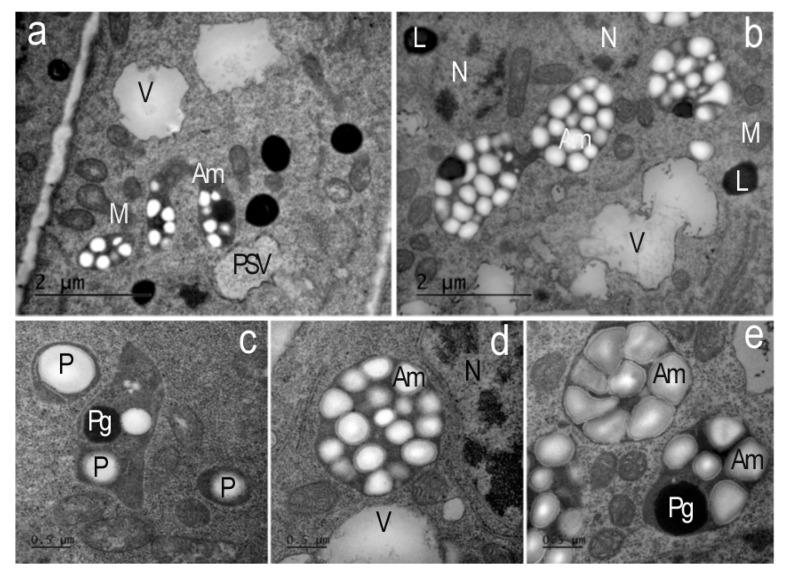
Amyloplasts of the endoderma and outer cortex of the apical meristem root cells of *Nicotiana tabacum* in the presence of AEDL. (**a**,**b**) In the cytoplasm of the outer cortex cells, plastids/typical amyloplasts with numerous starch grains and plastoglobuli are found. V—vacuoles; PSV—protein storage vacuole; L—lipid droplets; M—mitochondria; N—nucleolus. (**a**,**b**) Bar ×25,000. (**c**) In the cells of the endoderma, plastids (P) with individual starch grains and plastoglobuli (Pg) are found. (**d**,**e**)—amyloplasts (Am); M—mitochondria; V—vacuoles; N—nucleolus. (**c**–**e**) Bar ×50,000.

**Figure 9 ijms-26-11028-f009:**
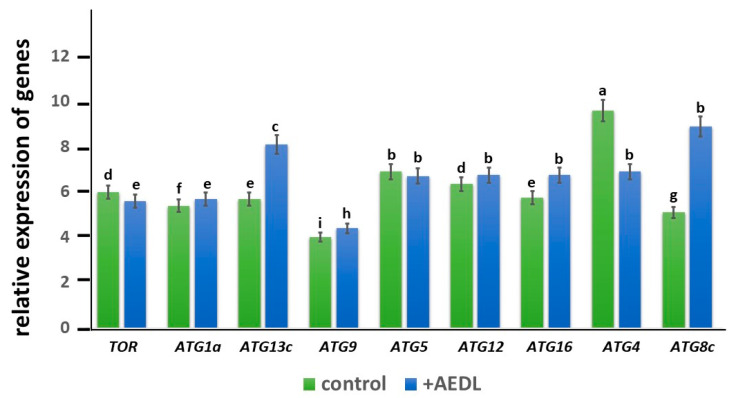
Expression of autophagy genes in the root of *Nicotiana tabacum.* The data were statistically significantly different from the above data; a–i indicate significant difference determined by one-way ANOVA (*p* < 0.05).

**Figure 10 ijms-26-11028-f010:**
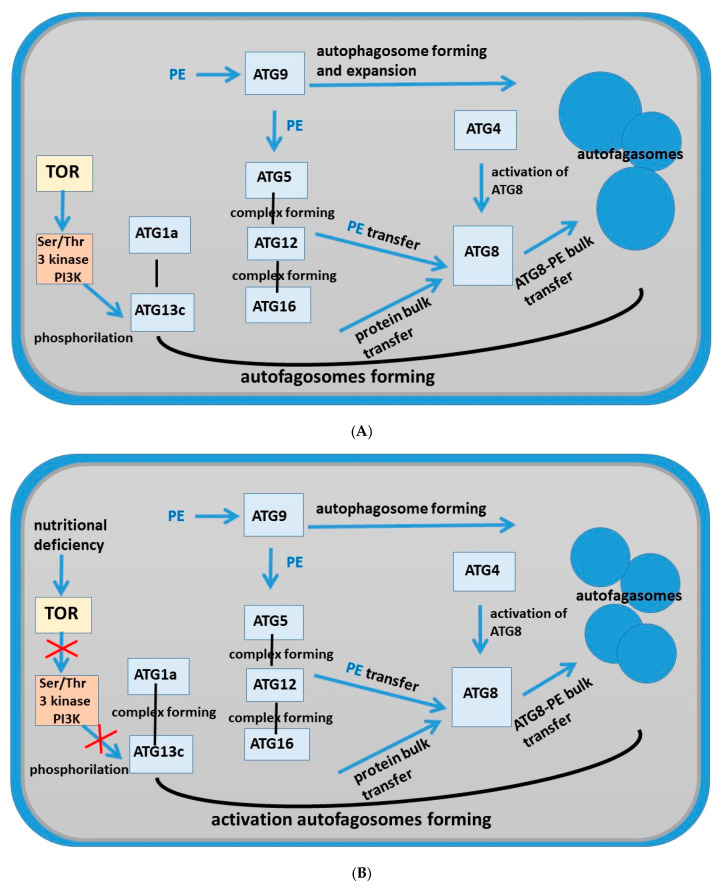
Schema of pathways of autophagy in root cells of *Nicotiana tabacum.* (**A**) Tobacco root cells grown under control conditions; (**B**) tobacco root cells grown in the presence of the AEDL peptide.

**Table 1 ijms-26-11028-t001:** Morphometric parameters of *Nicotiana tabacum* growth under different condition. Weight—total weight of root and shoot. Data are expressed as mean ± standard deviation (SD; *n* = 30), and a and b indicate significant difference determined by one-way ANOVA (*p* < 0.05).

**Variant**	**Root Length, cm**	**Shoot Height, cm**	**Crude Weight, g**	**Dry Weight, g**
Control	5.5 ± 0.27 b	3.7 ± 0.18 b	1.5 ± 0.07 b	0.41 ± 0.02 b
+AEDL	5.8 ± 0.29 a	4.8 ± 0.24 a	2.5 ± 0.12 a	0.58 ± 0.03 a

**Table 2 ijms-26-11028-t002:** DNA breaks in the nuclei of 1000 apical meristem cells of the root tips. Data are expressed as mean ± standard deviation (SD; *n* = 3), and a and b indicate significant difference determined by one-way ANOVA (*p* < 0.05).

Variant	Number of Cells with DNA Breaks	Number of Living Cells	% Dead Cells
Control	90 ± 5 b	910 ± 46 a	9%
+AEDL	140 ± 7 a	860 ± 43 b	14%

**Table 3 ijms-26-11028-t003:** Cytochrome c in mitochondria and in the cytoplasm of root meristem cells/in 1000 cells. Data are expressed as mean ± standard deviation (SD; *n* = 3), and a and b indicate significant difference determined by one-way ANOVA (*p* < 0.05).

Variant	Number of Cells Containing Cytochrome C in Mitochondria	Number of Cells Containing Cytochrome C in Cytoplasm	% Cells Containing Cytochrome C in Cytoplasm
Control	930 ± 46 a	70 ± 4 b	7%
+AEDL	820 ± 41 b	180 ± 9 a	18%

## Data Availability

The original contributions presented in this study are included in the article. Further inquiries can be directed to the corresponding author.

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
