# Peer review of "Peptide AEDL Activates Metabolism and Autophagy in Root Cells of Nicotiana tabacum"

_ijms, 2025, doi:10.3390/ijms262211028_

Round 1
Reviewer 1 Report
Comments and Suggestions for Authors
The effects of the short peptide AlaGluAspLeu (AEDL) on root development and the autophagy pathway in tobacco (Nicotiana tabacum) was investigated in this manuscript. By combining morphological observations, immunodetection, TUNEL assay, transmission electron microscopy (TEM), and autophagy-related gene expression analysis, it explores the potential mechanisms through which AEDL enhances plant development and metabolic activities, providing insights of the role of AEDL peptide in plant development and autophagy regulation.
Comments and suggestions:
1. In the abstract, "Nicotiana tabacum" should be italicized; "10-7 M" should be written as "10-7 M"; In Figure 1,higher resolution images should be provided with scale bars; In Tables 1, 2, and 3, variance significance analysis should be conducted and labeled in these tables;
2. Figures 2 and 3 should be combined into one figure, to enhance the comparison between them; and indicatingwhat the white arrows refer to;
3. Similarly, Figures 5 and 6should be combined into one figure. In Figure 6, what are F and G? What do A, Am, and PB represent?They should be indicated in the figure legends;
4. There is an issue with “Table 3. Cytochrome c in mitochondria and in the cytoplasm of root meristem cells/in 1000 cells.”,the indicating meaningin the main text seems to be contradict with the data in the table;
5. In Figure 7, the location of the phagophore needs to be pointed out. What does each of the letters represent? Please carefully check all images in Figures 1-9 of the manuscript, and indicate all letters in the figure captions;
6. In Figure 10,variance significance analysis need to be conducted, and significant differences should be marked in the figure;
7. In Experimental Methods"4.5. Immunodetection of ATG8 Antibodies", only a subtitle has been provided without corresponding experimental method content; In "4.6.. TEM Analysis", there is an extra dot; please check all the manuscript carefully;
8. In line 545, "p<0.05" should be italicizedas "p".
Author Response
Thank you for your careful reading of the manuscript and valuable comments.
The effects of the short peptide AlaGluAspLeu (AEDL) on root development and the autophagy pathway in tobacco (Nicotiana tabacum) was investigated in this manuscript. By combining morphological observations, immunodetection, TUNEL assay, transmission electron microscopy (TEM), and autophagy-related gene expression analysis, it explores the potential mechanisms through which AEDL enhances plant development and metabolic activities, providing insights of the role of AEDL peptide in plant development and autophagy regulation.
Comments and suggestions:
- In the abstract, "Nicotiana tabacum" should be italicized; "10-7 M" should be written as "10-7 M"; In Figure 1,higher resolution images should be provided with scale bars; In Tables 1, 2, and 3, variance significance analysis should be conducted and labeled in these tables;
Reply: Thanks for your comments. We've fixed it.
- Figures 2 and 3 should be combined into one figure, to enhance the comparison between them; and indicatingwhat the white arrows refer to;
Replay: Thanks for your comments. We've added scale bars to Figures 2 and 3, and the white arrows have been indicated in the captions for Figure 3. We don't think combining the figures is practical.
- Similarly, Figures 5 and 6should be combined into one figure. In Figure 6, what are F and G? What do A, Am, and PB represent?They should be indicated in the figure legends;
Reply: Thanks for the recommendation. We've combined figures 5 and 6. The figure captions have been edited.
- There is an issue with “Table 3. Cytochrome c in mitochondria and in the cytoplasm of root meristem cells/in 1000 cells.”,the indicating meaningin the main text seems to be contradict with the data in the table;
Reply: Sorry, there was a mistake. We've fixed it.
- In Figure 7, the location of the phagophore needs to be pointed out. What does each of the letters represent? Please carefully check all images in Figures 1-9 of the manuscript, and indicate all letters in the figure captions;
Reply: The figure captions have been edited.
Thank you for your comment. All letter designations have been deciphered in all figures and included in the corresponding captions.
- In Figure 10,variance significance analysis need to be conducted, and significant differences should be marked in the figure;
Reply: Thanks for the comment. Significant differences have been made.
- In Experimental Methods"4.5. Immunodetection of ATG8 Antibodies", only a subtitle has been provided without corresponding experimental method content; In "4.6.. TEM Analysis", there is an extra dot; please check all the manuscript carefully;
Reply: Sorry, there was an error. We've fixed it.
- In line 545, "p<0.05" should be italicizedas "p".
Reply:. We've fixed it.

Reviewer 2 Report
Comments and Suggestions for Authors
The manuscript is devoted to study of the ultrastructure of Nicotiana tabacum root cells and molecular processes in the presence of the short peptide AlaGluAspLeu (AEDL). The distribution of the autophagy marker protein Atg8 in the cytoplasm of outer and inner cortex meristem cells of the tobacco root tip was studied.
The topic of the work is of interest and the Authors made a lot of work (experimental part). After reading the work, there are several regarding and suggestions that should be taken into account before accepting the work for publication:
- There are several unclear points in the description of the plant growth methodology. What light sources were used in the experiment? What was the photon flux density during photosynthesis? Experimental variants should be indicated in the description. Lines 482-483: "The experiment was carried out for 4 weeks 482 in 4 replicates". What number of plants were in each replicate? What was the total sample size of plants for the morphometric analysis? What does the number n=3 in the caption to Table 1 mean? Specify which morphometric parameters were taken into account in the experiment?
- In the "materials and methods" section there is no description of sub-sections 4.5. Immunodetection of ATG8 antibodies.
- Lines 482-483: "Values are presented as means ± standard deviations of triplicate biological replicates". Is triplicate biological replication sufficient for statistical analysis of the morphometric traits studied?
- Please correct the captions for tables 2 and 3, which repeat the caption for table 1.
- In Tables and the captions, statistically significant differences, the statistical test used, and the significance level should be indicated.
- It is necessary to decipher the abbreviations ER, M (Figure 4), PB, Am (Figure 6), Nu, A, F, M, V, PSV, CW (Figure 7), Nu, A, P, V, CW (Figure 8), all for Figure 9. Label photos F and G for Figure 6. The abbreviations PSV, Am, LV and A are in the caption to Figure 6 are not indicated in the pictures.
- “A significant increase in root system and aerial part size was demonstrated in Nicotiana tabacum grown in the presence of the AEDL peptide (Fig. 1)” (lines 302-303). Statistical confirmation of the results is not provided.
- The "discussion" section needs to be edited as it contains repetitions of the same text fragments, for example, in lines 267–269, 272–274, 265–266, and 274–276.
- In the "discussion" section, some sentences require reference(s), such as lines 346-349, 472-474. Review this throughout the text.
- Lines 548-549: "Nicotiana tabacum grown in the presence of AEDL peptide exhibited larger leaf and stem sizes...". Data and statistical analysis for traits "leaf size" and "stem size" are not presented in the main text.
Author Response
Thank you for your careful review of the manuscript and valuable comments.
The manuscript is devoted to study of the ultrastructure of Nicotiana tabacum root cells and molecular processes in the presence of the short peptide AlaGluAspLeu (AEDL). The distribution of the autophagy marker protein Atg8 in the cytoplasm of outer and inner cortex meristem cells of the tobacco root tip was studied.
The topic of the work is of interest and the Authors made a lot of work (experimental part). After reading the work, there are several regarding and suggestions that should be taken into account before accepting the work for publication:
There are several unclear points in the description of the plant growth methodology. What light sources were used in the experiment? What was the photon flux density during photosynthesis? Experimental variants should be indicated in the description. Lines 482-483: "The experiment was carried out for 4 weeks 482 in 4 replicates". What number of plants were in each replicate? What was the total sample size of plants for the morphometric analysis? What does the number n=3 in the caption to Table 1 mean? Specify which morphometric parameters were taken into account in the experiment?
Reply: Thanks for the comment, we've corrected it to 3 repetitions.
In the "materials and methods" section there is no description of sub-sections 4.5. Immunodetection of ATG8 antibodies.
Reply: Sorry, we've added a missing immunodetection method.
Lines 482-483: "Values are presented as means ± standard deviations of triplicate biological replicates". Is triplicate biological replication sufficient for statistical analysis of the morphometric traits studied?
Reply: Thank you for your question. Three repeats are sufficient for reliable cytological studies.
Please correct the captions for tables 2 and 3, which repeat the caption for table 1.
Reply: Thanks for the comment. We've removed the unnecessary information.
In Tables and the captions, statistically significant differences, the statistical test used, and the significance level should be indicated.
Reply: In the tables we used only SD
It is necessary to decipher the abbreviations ER, M (Figure 4), PB, Am (Figure 6), Nu, A, F, M, V, PSV, CW (Figure 7), Nu, A, P, V, CW (Figure 8), all for Figure 9. Label photos F and G for Figure 6. The abbreviations PSV, Am, LV and A are in the caption to Figure 6 are not indicated in the pictures.
Reply: Thank you for your valuable feedback. All figure captions are provided in accordance with all notations.
“A significant increase in root system and aerial part size was demonstrated in Nicotiana tabacum grown in the presence of the AEDL peptide (Fig. 1)” (lines 302-303). Statistical confirmation of the results is not provided.
Reply: Statistical analysis of morphometric parameters was provided in earlier published articles. These values are not discussed in this manuscript.
The "discussion" section needs to be edited as it contains repetitions of the same text fragments, for example, in lines 267–269, 272–274, 265–266, and 274–276.
In the "discussion" section, some sentences require reference(s), such as lines 346-349, 472-474. Review this throughout the text.
Reply: Thank you for your comment. We have edited the text.
Lines 548-549: "Nicotiana tabacum grown in the presence of AEDL peptide exhibited larger leaf and stem sizes...". Data and statistical analysis for traits "leaf size" and "stem size" are not presented in the main text.
Reply: Data and statistical analysis for the leaf size and stem size traits have been presented and discussed in earlier publications.

Reviewer 3 Report
Comments and Suggestions for Authors
The manuscript concerns the influence of regulatory peptide AlaGluAspLeu (AEDL) on the growth of Nicotiana tabacum. The study is interesting and provides new valuable data on autophagy.
The background and bibliography are appropriate, the main aim is stated in the end of the introduction. However, I would suggest some modifications of the Abstract (details below).
The methods are well selected, but not all of them have been properly described in Material and Methods section (for example there is no “Immunodetection of ATG8 antibodies” chapter). The presentation of the results is clear, however, some captions to figures need corrections (listed below). The discussion is sufficient.
Detailed comments:
Abstract should be rewritten - the aim of the study should be clearly stated, and the applied methods should be briefly mentioned.
Please correct the captions to figure 2:
Figure 2. ”Immunodetection protein Atg8 into control macerated cells of the apical meristem tobacco 122 root tips. Meristem root cells of inner (endoderma) cortex (А) and outer cortex (В).“ into:
Figure 2. Immunodetection of protein Atg8 in control macerated cells of the apical meristem tobacco root tips. A- Meristem root cells of inner (endoderma) cortex, B -outer cortex.
And similarly correct the caption of Figure 3.
Correct “amiloplasts” into amyloplasts in the caption to Figure 9.
Correct the caption of the Figure 11 - Schema of pathways (or: Pathway schema) instead of “Schema pathways”.
In Material and methods:
Line 480. “AEDL was added to the medium in parallel”. Please correct this part, it is not clear! What means “in parallel”? It should be clearly stated what was the control, and what was the
Line 519. The chapter entitled “Immunodetection of ATG8 antibodies” is missing.
Conclusion
LIne 548 …”had larger sizes of both leaves and stems (above-ground parts), as well as the root system” – better to write that had larger size and bigger biomass (sometimes elongation is not connected to increase in biomass).
Author Response
Thank you for your careful reading of our manuscript and valuable comments.
The manuscript concerns the influence of regulatory peptide AlaGluAspLeu (AEDL) on the growth of Nicotiana tabacum. The study is interesting and provides new valuable data on autophagy.
The background and bibliography are appropriate, the main aim is stated in the end of the introduction. However, I would suggest some modifications of the Abstract (details below).
The methods are well selected, but not all of them have been properly described in Material and Methods section (for example there is no “Immunodetection of ATG8 antibodies” chapter). The presentation of the results is clear, however, some captions to figures need corrections (listed below). The discussion is sufficient.
Detailed comments:
Abstract should be rewritten - the aim of the study should be clearly stated, and the applied methods should be briefly mentioned.
Please correct the captions to figure 2:
Figure 2. ”Immunodetection protein Atg8 into control macerated cells of the apical meristem tobacco 122 root tips. Meristem root cells of inner (endoderma) cortex (А) and outer cortex (В).“ into:
Figure 2. Immunodetection of protein Atg8 in control macerated cells of the apical meristem tobacco root tips. A- Meristem root cells of inner (endoderma) cortex, B -outer cortex.
Reply: Thank you for your comment. We've corrected the figure captions according to your recommendations.
And similarly correct the caption of Figure 3.
Correct “amiloplasts” into amyloplasts in the caption to Figure 9.
Reply: Thanks for the comment. We've fixed the error.
Correct the caption of the Figure 11 - Schema of pathways (or: Pathway schema) instead of “Schema pathways”.
Reply: Thanks for the comment. We've fixed the error.
In Material and methods:
Line 480. “AEDL was added to the medium in parallel”. Please correct this part, it is not clear! What means “in parallel”? It should be clearly stated what was the control, and what was the
Reply: Thanks for the comment. Thanks for your comment. We've edited the sentence.
Line 519. The chapter entitled “Immunodetection of ATG8 antibodies” is missing.
Sorry, we've added a missing immunodetection method.
Conclusion
LIne 548 …”had larger sizes of both leaves and stems (above-ground parts), as well as the root system” – better to write that had larger size and bigger biomass (sometimes elongation is not connected to increase in biomass).
Reply: Thanks for the comment. Thanks for your comment. We've edited the sentence.

Round 2
Reviewer 1 Report
Comments and Suggestions for Authors
Many of the issues were resolved, however, there are still some needed to be corrected. The authors should not just response to the comments without correcting the errors in the context of the manuscript.
Comments and suggestions:
1. In the abstract, "Nicotiana tabacum " should be italicized (line 14-15); "10-7 M" should be written as "10-7 M" (line 27);
2. In Figure 1, higher resolution images should be provided;
3. In Tables 1, 2, and 3, variance significance analysis should be conducted and labeled in these tables, similar to the labeling in figure 9;
4. Again, in Tables 3, the third column “% cell containing cytochrome c in mitochondria/cytoplasm” seems to be “% cell containing cytochrome c in cytoplasm”. Please correct it.
5. In Figures 4, no scale bar was found in the panels, please add the scale bars.
6. As there are multiple same letters marked in the same figure, including “A” “F”, it is suggested to change the panel marked letters from uppercase letters into lowercase letters, for instance, from panel “A” “B” “C” “D” “E” “F” “G” into panel “a” “b” “c” “d” “e” “f” “g”, and unify all the indicated letters throughout the manuscript.
Author Response
Many of the issues were resolved, however, there are still some needed to be corrected. The authors should not just response to the comments without correcting the errors in the context of the manuscript.
Comments and suggestions:
- In the abstract, "Nicotiana tabacum " should be italicized (line 14-15); "10-7 M" should be written as "10-7 M" (line 27);
Reply: Sorry for the errors. We've fixed them.
- In Figure 1, higher resolution images should be provided;
Reply: Sorry, unfortunately we couldn't improve it.
- In Tables 1, 2, and 3, variance significance analysis should be conducted and labeled in these tables, similar to the labeling in figure 9;
Reply: Corrected
- Again, in Tables 3, the third column “% cell containing cytochrome c in mitochondria/cytoplasm” seems to be “% cell containing cytochrome c in cytoplasm”. Please correct it.
Reply: Sorry for the errors. We've fixed them.
- In Figures 4, no scale bar was found in the panels, please add the scale bars.
Reply: We inserted scale rulers.
- As there are multiple same letters marked in the same figure, including “A” “F”, it is suggested to change the panel marked letters from uppercase letters into lowercase letters, for instance, from panel “A” “B” “C” “D” “E” “F” “G” into panel “a” “b” “c” “d” “e” “f” “g”, and unify all the indicated letters throughout the manuscript.
Reply: We have changed the letter designations in accordance with your comments.

Reviewer 3 Report
Comments and Suggestions for Authors
The Authors have improved the manuscript. I have only two requests:
Line 484. This description still does not indicate what was a control. Please correct it, for example: “Tobacco seeds (Nicotiana tabacum L.) of the Samsun variety were placed in test tubes containing Murashige-Skoog (MS) medium without hormones (control samples), the medium of test samples was supplemented with 10-7M AEDL.
Line 560. “It shown that Nicotiana tabacum grown in the presence of the AEDL peptide had that had larger size and bigger biomass.” Please delete “that had” (…Nicotiana tabacum grown in the presence of the AEDL peptide had larger size and bigger biomass)
Author Response
The Authors have improved the manuscript. I have only two requests:
Line 484. This description still does not indicate what was a control. Please correct it, for example: “Tobacco seeds (Nicotiana tabacum L.) of the Samsun variety were placed in test tubes containing Murashige-Skoog (MS) medium without hormones (control samples), the medium of test samples was supplemented with 10-7M AEDL.
Reply: Thanks, fixed it
Line 560. “It shown that Nicotiana tabacum grown in the presence of the AEDL peptide had that had larger size and bigger biomass.” Please delete “that had” (…Nicotiana tabacum grown in the presence of the AEDL peptide had larger size and bigger biomass)
Reply: Thanks, fixed it
